# Bacterial Motility and Its Role in Skin and Wound Infections

**DOI:** 10.3390/ijms24021707

**Published:** 2023-01-15

**Authors:** Katarzyna Zegadło, Monika Gieroń, Paulina Żarnowiec, Katarzyna Durlik-Popińska, Beata Kręcisz, Wiesław Kaca, Grzegorz Czerwonka

**Affiliations:** 1Department of Microbiology, Institute of Biology, Faculty of Natural Sciences, Jan Kochanowski University, 25-406 Kielce, Poland; 2Department of Dermatology, Cosmetology and Aesthetic Surgery, The Institute of Medical Sciences, Medical College, Jan Kochanowski University, 25-317 Kielce, Poland

**Keywords:** swimming, swarming, twitching, gliding, sliding, motility, bacteria, skin, wound

## Abstract

Skin and wound infections are serious medical problems, and the diversity of bacteria makes such infections difficult to treat. Bacteria possess many virulence factors, among which motility plays a key role in skin infections. This feature allows for movement over the skin surface and relocation into the wound. The aim of this paper is to review the type of bacterial movement and to indicate the underlying mechanisms than can serve as a target for developing or modifying antibacterial therapies applied in wound infection treatment. Five types of bacterial movement are distinguished: appendage-dependent (swimming, swarming, and twitching) and appendage-independent (gliding and sliding). All of them allow bacteria to relocate and aid bacteria during infection. Swimming motility allows bacteria to spread from ‘persister cells’ in biofilm microcolonies and colonise other tissues. Twitching motility enables bacteria to press through the tissues during infection, whereas sliding motility allows cocci (defined as non-motile) to migrate over surfaces. Bacteria during swarming display greater resistance to antimicrobials. Molecular motors generating the focal adhesion complexes in the bacterial cell leaflet generate a ‘wave’, which pushes bacterial cells lacking appendages, thereby enabling movement. Here, we present the five main types of bacterial motility, their molecular mechanisms, and examples of bacteria that utilise them. Bacterial migration mechanisms can be considered not only as a virulence factor but also as a target for antibacterial therapy.

## 1. Introduction

The life and proper functioning of the human body would not be possible without having extensive bacterial flora, which is present not only in the digestive and reproductive systems but also on the skin, which primarily has protective functions and is responsible for maintaining the proper homeostasis of the body. The skin microflora, also termed the skin microbiome, varies according to the sex and age of the host and is topographically diverse due to anatomical differences in the human body. The development of the microbiome on the skin in different bodily regions mainly depends on oxygen availability, the presence of sebaceous glands, humidity, and the temperature of the individual zones. It includes not only bacteria but also viruses, fungi, and bacteriophages [1,2]. The basic barrier, which is the skin, protects against mechanical and thermal factors, harmful radiation, and infections. The disruption of skin homeostasis, in combination with pathogen appearance, may lead to the development of various skin infections. Infection development is fostered by various factors, which include being immunocompromised, vascular insufficiency, poor lymphatic or venous drainage, sensory neuropathies, diabetes, obesity, poor hygiene, and surgical procedures [3]. Infection occurrence is also influenced by the virulence of microorganisms and the degree of their presence. The simplest division of bacterial infections is acute and chronic infections. Pathogenic bacteria cause a wide variety of acute skin infections with varied clinical appearances depending on the aetiological factor, location, and immune status of the host. The most common acute infections are staphylococcal, streptococcal, and mixed staphylococcal–streptococcal infections [4,5]. In addition, other species also cause skin infections, including *Borrelia* spp., *Pseudomonas aeruginosa*, *Corynebacterium* spp., *Actinomyces* spp., *Bacillus anthracis*, *Listeria monocytogenes*, *Nocardia* spp., *Erysipelothrix rhusiopathiae*, *Bartonella* spp., *Klebsiella pneumoniae*, *Klebsiella rhinoscleromatis*, *Yersinia* spp., *Clostridium* spp., *Francisella tularensis*, *Brucellae* spp., and *Mycobacterium* spp. [6]. Moreover, bacteria that would not be pathogenic with an undamaged skin barrier can penetrate through damaged skin and become virulent. In general, most acute infections are superficial and relatively easy to treat with appropriate antibiotics [7], whereas chronic infections, in particular chronic wound infections, often pose many difficulties in effective treatment, and in the era of multi-drug resistance, they are a major social and economic problem [8]. The most common aetiological factors in chronic wounds are *Staphylococcus aureus*, *P. aeruginosa*, *Acinetobacter baumannii*, *Proteus mirabilis*, *Escherichia coli*, and *Corynebacterium* spp. [9,10,11]. Among them, there are many antibiotic-resistant species included in the “ESKAPE” cluster composed of *Enterococcus faecium*,* S. aureus*, *K. pneumoniae*, *A. baumannii*,* P. aeruginosa*, and *Enterobacter* species [12]. Due to the increasing drug resistance of bacteria responsible for skin infections, various studies have concentrated on the treatment of infections of wounds using various methods to reduce virulence factors [13,14,15]. Some virulence factors include biofilm formation, exotoxins and endotoxins, proteases, quorum sensing, and motility [14,16].

Microbes enter the wound in three main ways: bacterial deposition from air or water, direct contact with infected material, and through the transfer of one’s own physiological flora [17]. There are four microbial states of the infected wound (Table 1).

The basic methods for treating chronic wounds are the use of antiseptics, a great variety of wound dressings, antibiotic therapy, compression therapy, and debridement. Other popular treatment options are hyperbaric oxygen therapy, negative pressure wound therapy, ultrasound, and electromagnetic therapy [20]. Hyperbaric oxygen therapy is widely in use for more than 20 years in diabetic foot ulcers, and this therapy may improve wound healing by stimulating angiogenesis. The mechanism of action of hyperbaric oxygen therapy on tissues has not yet been fully elucidated [21]. Another promising method for skin and wound treatment is negative pressure wound therapy, where the use of a vacuum reduces oedema and infection and increases local blood flow, which promotes healing [22]. This method is widely used as an alternative to surgery for a wide range of wounds [23]. Ultrasound therapy is the use of mechanical energy at high frequency (between 1 and 3 MHz) to induce thermal energy into musculoskeletal tissues and low frequency (between 22.5 and 40 kHz) to generate the cavitation effect, which, if delivered in waves, debride necrotic tissue from wound surface [24]. Interestingly, skin wounds generate large and endogenous electric fields termed “current of injury” [25]. This electrical skin activity is involved in the process of wound healing, so it has led to the hypothesis that applied electrical stimulation may support the healing of chronic wounds by imitating the natural electrical current [26]. Several electrical stimulation methods are available, including pulsed current, direct current, and rhythmic electric frequency. However, the best strategy of electrical stimulation is still unachieved [26]. Unfortunately, the above-mentioned methods frequently fail; thus, chronic wound treatment is difficult, long-lasting, and sometimes ineffective. As the problem of chronic wounds affects approximately 1–2% of the global population, innovative and more effective methods for treating hard-to-heal wounds are constantly being sought [27,28].

Consequently, it is imperative to study the bacterial colonisation of poorly healing wounds and assign them to specific taxonomic units, together with analysing their numerous functions and relationships. These factors underlie their unsurpassable ability to evolve and adapt to their environment, which has led to the rising issue of antibiotic resistance. Understanding this will bring us closer to a future with increasingly effective and less invasive therapies and improvement in the entire process of patient convalescence. Most wound-infecting bacteria are motile. Bacterial motility plays a key role in skin and wound infection, which leads to hindering therapy and full recovery. The molecular mechanisms underlying bacterial motility may be an attractive target for new therapies being developed, where the inhibition of motility will not only reduce the spread of infection but also facilitate bacterial eradication. Here, we present an overview of the types of bacterial movement along with a description of their molecular machinery, to indicate them as a potential target of interest.

## 2. Types of Bacterial Movement

One of the main characteristics of microbes adapting to and surviving in the external environment is how they move and grow, whether in liquid, semi-liquid, or solid environments. Notably, not all bacteria are capable of active movement. In most cases, the flagella enable motion. Bacterial cells have flagella in variable quantities and locations, which is one of the features that enable their classification. However, different movement phenomena, such as the result of type IV pili (T4P), are still being investigated. In 1972, the different types of microbial translocation were characterised and named [29]. Five of the original terms (swimming, swarming, twitching, gliding, and sliding) are still widely used [30]. Differences in bacterial types of movement are shown in Figure 1.

### 2.1. Swimming Motility

Swimming motility is one of the basic skills in the life cycle of bacteria, giving them an advantage in surviving unfavourable environmental conditions [32]. Swimming motility is an individual cell movement that uses flagella rotation to move through aqueous environments [33,34]. Flagellated bacteria typically swim in a series of more or less straight runs interrupted by short reorientations [35]. Each flagellum is rotated by a motor that is embedded in the cell membrane. Flagella are highly complex bacterial organelles that are unusually well conserved among diverse bacterial species. Over 60 structural and regulatory proteins are involved in flagellum synthesis and function [36]. There are three elements in the structure of flagella found in prokaryotes: the fibre, the hook, and the basal body. A cell may have one, two, or more flagella on the front or back of the cell, over part of it, or covering its entire surface. The outer part of the cilia consists of the flagellin protein, of which there are approximately 20,000 subunits in a single cilium. The flagellum is anchored in the bacterial cell by a basic body consisting of the L, P, S, M, and C rings surrounding the cylindrical part. Gram-positive bacteria do not have the two outer L and P rings, which distinguishes them from Gram-negative bacteria with five rings [37]. The flagella fibre is connected to the basic body with a flexible section called a hook, the length of which is approximately 55 nm [38]. The direction and regulation of flagellar rotation enable bacteria to move in chemical gradients, termed chemotaxis [35,39]. This complex behaviour begins in cell membrane chemoreceptors, which detect chemical compounds and respond to them by changing their conformation. *E. coli* has five such receptors (Tar, Tsr, Tap, Trg, and Aer), which are arranged in chemoreceptor clusters, together with two cytoplasmic proteins, the adaptor CheW and the histidine autokinase CheA [40]. Motility regulation by the chemotaxis system is well investigated for a variety of bacterial species. Chemical gradients are sensed by chemoreceptors, which trigger the autophosphorylation of the cytoplasmic CheA, which forms a complex with the receptor through the coupling protein CheW [33,41]. In response to these changes, CheA transfers its phosphate group to CheY, a diffusible cytoplasmic response regulator that interacts with the flagellar motor and leads to a modulation of motility characteristics [33,41]. This signalling core is highly conserved among all chemotaxis pathways [42].

Although only a fraction of bacteria associated with animal hosts are motile, flagellar motility and chemotaxis are important for the successful colonisation and virulence of many pathogens, for example, gastrointestinal *Campylobacter jejuni*, *Salmonella enterica* serovar Typhimurium, *Helicobacter pylori*, and *Vibrio cholerae* [41]. In several cases, it has also been shown that the same flagellin, the protein subunit comprising the bacterial flagellum, can be a major driver of inflammation [35].

Motility might have several functions in the host–microbe interactions. For example, in the early stages of biofilm formation, planktonic bacteria swim close to the surface by rotating their flagella and attaching to the surface using their pili [43]. The most common opportunistic pathogen in wound infections is *P. aeruginosa*. Many studies have demonstrated that *P. aeruginosa* biofilms are the key factors for exacerbating the skin inflammatory response and its resistance to antimicrobial agents [13,36].

Bacteria are inextricably associated with wound healing. However, the latest research indicates that they are not always associated with infection and worse wound healing. It has been reported that the skin’s natural microbiome can contribute to more rapid wound healing. Several studies were performed with mice infected with staphylococci. Interestingly, *S. aureus* was found to be the best healing inducer.

Sometimes, swimming bacteria are used by non-flagellated bacteria that do not have the capacity to independently translocate with this mechanism. The genus *Staphylococcus*, for example, is classically considered non-motile in fluid environments due to a lack of flagella. Despite their motility limitations, staphylococcal species effectively reach and thrive in their preferred ecological niches. Data suggested that *S. aureus* has acquired, through *P. aeruginosa*, an increased capacity to travel longer distances, allowing it to colonise niches that are relatively inaccessible in the absence of swimming carrier bacteria, which may be due to the hitchhiking of *S. aureus* on *P. aeruginosa*. It was also observed that *P. aeruginosa* can carry another staphylococcal cargo, *Staphylococcus epidermidis*, which may also be important in wound infection [44].

### 2.2. Swarming Motility

Swarming motility is mainly based on the differentiation of vegetative cells into swarming cells with a large number of flagella, which can migrate rapidly in a coordinated manner over solid surfaces. Bacteria display different phenotypes during swarming, but not all of them are equal: swarming *P. aeruginosa* do not develop typical swarmer cell phenotypes (elongated, hyper-flagellated cells) similar to other swarming bacteria, and individual cells swarming in rafts also reveal great variance and the lack of a highly differentiated swarmer cell phenotype [45]. Swarming phenotypes vary not only within a species but also within a strain, whereby the different *P. aeruginosa* strains grown on identical medium conditions will display different swarming patterns [46]. A swarm of migrating bacteria moves forward and traps a water reservoir, and in this moist region, individual cell speed is comparable to swimming speeds in bulk liquid, typically in the order of 20 μm/sec [47]. Bacterial swarming plays a crucial role in many pathogen–host interactions, and is considered an important virulence factor. The differentiation during swarming into the hyper-flagellated elongated cell of *P. mirabilis* is coupled to the ability of this bacterium to enter host cells, with the manifestation of upregulated virulence protein expression (haemolysin, urease, and protease) [48,49].

The stimuli that are necessary to control this movement are responsive to cell density, surface contact, and physiological signals. The swarming motility needs signals from the quorum sensing system and the cyclic di-GMP network that regulates the transitions between motile and biofilm modes across many bacterial species [50]. These signals activate flagella biogenesis via the main flagellar operon *flhDC*, which is the main point of the regulatory network of cell differentiation and migration and is crucial for swarming motility [48,51]. Surface sensing occurs in two proposed ways, through the inhibition of flagellar rotation and/or through the detection of the O antigen contact of lipopolysaccharide (LPS) with a solid surface [51] in Gram-negative bacteria. In addition, LPS is suggested to act as an osmolarity agent and facilitate swarming [52].

The swarming phenomenon is characterised by bacterial growth in the form of zones with clearly delimited darker peripheral circles on an agar medium. In a typical swarming, after contact with a surface, microbial cells are able to extend up to 40 times and increase the number of flagella. In the following stage, cells divide into short cells that are unable to move. The cycles of cell elongation, movement, and division are repeated constantly, resulting in a characteristic “pattern” of their growth on the substrate, in which the darker rings reflect the periods of bacterial division. Changes in cell length and shape result from changes in the hardness of the surface on which the cells grow, and the presence of LPS, which participates in the regulation of osmolarity, helps them overcome the environmental barriers that inhibit bacterial mobility. Among others, *E. coli*, *Salmonella* spp., *P. mirabilis*, *Bacillus subtilis*, and *P. aeruginosa* are able to migrate in this way. Compared with swimming, chemotaxis is unnecessary here, and the oscillating motion is mainly based on the flagellar drive and the mechanical interactions taking place [53]. The swarming motility is also closely associated with the resistance of bacterial cells, which are antibiotic-sensitive, providing cells with greater availability of nutrients and competitive advantage due to secreted surfactants [48]. For example, the multi-resistant phenotype of *P. aeruginosa* is closely associated with swarming growth and has a transient form [54]. Currently, research on this motility is focused on obtaining more detailed information about their lipids, proteins, and enzymes, as well as their relationships, the metabolic pathways involved, and methods of regulation at the molecular level [55].

### 2.3. Twitching Motility

When a single cell adheres to a solid surface, it may initiate a type of surface-associated motility termed twitching. This is a movement across abiotic and biotic surfaces such as glass, plastic, or skin tissues, which is mediated by hair-like, multi-subunit compounds T4P [56,57,58]. T4P machinery is a subclass of the type IV filament superfamily closely associated with the type II secretion system (T2SS) of Gram-negative bacteria as well as the competence pilus of Gram-positive bacteria and the archaellum in Archaea [59,60]. T4P are widespread in bacteria, including important human pathogens such as *P. aeruginosa*, *Neisseria gonorrhoeae*, and *V. cholerae* [61], and play an important role in, for example, cell–cell adherence, cell-surface adherence, DNA transfer in the environment, predation, surface sensing, and motility [31].

The mechanism of twitching motility is based on cycles of rapid extension, adhesion to the surface, and the retraction of T4P at rates of approximately 1 μm/s [31]. Although the T4P assembly system is conserved between bacterial species, the nomenclature of T4P proteins is very heterogeneous [62]. Ten conserved proteins have been described for *Myxococcus xanthus*, as revealed by cryo-electron tomography. Nine of these form an envelope-spanning complex termed the basal body [63]. On the cytoplasmic side, the basal body consists of a cytoplasmic dome (PilC) separating two antagonistically acting ATPases (PilT and PilB) forming a disk-like structure and acting as a motor, providing energy for pilus assembly. The cytoplasmic disk is surrounded by a cytoplasmic ring (PilM) and is interconnected with two distinct periplasmic rings, the lower periplasmic ring (PilN and PilO) and the mid periplasmic ring (PilP), and with one ring immediately below the OM (TsaP) surrounding the channel (PilQ) of the OM pore. The final protein PilA is the major N-terminal hydrophobic α-helix protein that forms the extracellular pilus fibre [63]. PilA is not the only protein that forms the extracellular fibre, because there are some minor pilins that form a priming complex and, therefore, localise at the pilus tip. Different minor pilins are involved in different biological functions, for example, aggregation via pilus–pilus interactions and the acquisition of external DNA and motility (the structure and function of minor pilins of T4P) [58,62,63]. Extracellular fibre filaments range from thin (6–9 nm in diameter) to long (several microns) and can expand to be many times longer than the cell [64].

Although the term twitching refers to fast incoherent movements, there are studies that show that single cells moving with T4P can move in one direction for over an hour, and individual T4P compounds generate a force of approximately 100–150 pN [31,65,66,67]. Moreover, the cell speed during twitching may vary from approximately 0.02 µm/s to 0.16 µm/s, which depends on environmental conditions, for example, the number of pili involved, the surface, oxygen conditions, and the chemotactic factor [33,40,41,43]. Studies indicated that twitching can be induced chemotactically and/or mechanotactically; in *P. aeruginosa*, this signalling is suggested in both cases to be attributed to the Chp operon [66,68].

Spherical bacterial forms distribute the T4P over the entire cell surface, and movement is coordinated through a tug-of-war mechanism, whereas rod-shaped bacteria have T4P located on the poles [31,67]. On the basis of a time-averaged tilt angle of the bacterial long axis with respect to the surface, and the slope of the mean-squared displacement curve of the bacterial trajectory, three types of twitching motility can be distinguished. Walking upright is a movement of a cell attached to the surface with one pole. The frequent alteration of their twitching directions results in a high angular speed movement. Wiggling is a movement when bacterial cells display bipolar attachment, and this normally generates low net translational motion. Crawling is a movement when bacteria also display bipolar attachment and tends to generate high net translational motion. In comparison to walking, both crawling and wiggling cells have low angular speeds [69].

Twitching is associated with tissue invasion and virulence [70]. A correlation between wound area and bacterial twitching has been demonstrated [16]. The clinical strains of *P. aeruginosa* isolated from burn wound patients possess T4P as the acute virulence factor required for the colonisation and establishment of burn wound infections [14]. Moreover, twitching promotes microcolonies and biofilm formation, which influences chronic wound infection development [71]. Data from the literature indicated that T4P mutations result in the formation of incorrect mature forms of biofilm. In addition, it has been suggested that twitching can be used by bacterial cells to climb on microcolonies formed by a subpopulation of immobile cells attached to the surface, creating a mushroom-like architecture of mature biofilm [72]. Negative pressure wound therapy may reduce the motility of *P. aeruginosa* and biofilm formation and thereby enhance wound healing [13]. Twitching also promotes the dispersal of biofilm cells, making the released bacteria susceptible to the action of antibacterial substances [73]. There are known anti-biofilm peptides stimulating twitching motility, including synthetic 1037 and natural LL-37. Low levels of iron stimulate bacterial twitching and biofilm dispersion. Lactoferrin increases twitching motility by limiting the iron level in the bacterial environment [73].

### 2.4. Gliding Motility

Gliding and sliding forms of motility are considered to be the ability to translocate on solid surfaces by a flagellum-independent mechanism of smooth motion along the long axes of rod-shaped bacteria [74]. The physical principles behind gliding motility involve no external appendage utilisation, and they remain poorly understood. Gliding motility is known as a horizontal translation of bacteria under zero net force [75]. On soft substrates, the thrust of bacterial cells arises from bacterial shape deformations, which result in a flow of slime along the bacterial length [75]. Gliding motility, as observed in *Myxococcus xanthus*, can be distinguished from A-motility, which allows individual cells to glide on a surface, and S-motility, dependent on the functional T4P movement of groups of bacterial cells that are in close proximity to each other [76]. In recent decades, a model for gliding motility was proposed where focal-adhesion complexes (FACs) that penetrate the cell envelope and are anchored on the cell to the substratum were believed to be responsible for this motility [77]. Unfortunately, this idea assumed that FACs would migrate along the cell, leading to cell wall reassembly, including peptidoglycan, which appeared unlikely to occur [78]. Other studies revealed that FACs are generated by molecular motors (AglR, AglQ, and AglS) located in the inner membrane of the cell and are proton-conducting channels similar to the MotAB flagellar stator complex and the transport channel TolQR. Moreover, further studies showed that the FACs could be considered a fluid aggregate formed by motors and cargo proteins that accumulate at a contact site with a solid surface, which generates a slight deformation of the cell envelope and subsequently generates the force that pushes the cell forward [74,79,80]. While gliding motility was intensely studied and characterised in *M. xanthus*, this type of motility was also reported for *Bacteroidetes* and the genus *Mycoplasma* [30], but the gliding mechanism in other bacteria may be different from that determined in myxococci [81].

### 2.5. Sliding Motility

Sliding motility is considered to be a passive form of bacterial movement on a solid surface, where the motion force mainly originates from the force of dividing cells pushing other cells forward. Sliding is promoted by several factors reducing the friction between cells and the substratum, including exopolysaccharides, hydrophobic proteins, and glycopeptides. This form of motility is considered to play an important role in surface colonisation by mycobacteria in the environment as well as in the host [82]. Sliding bacteria were classified by the current knowledge of different sliding mechanisms. The first group of sliding bacteria requires only the pushing force of cell division and secreted surfactant. This first group of sliding bacteria includes *Serratia marcescens*, *P. aeruginosa*, *Pseudomonas syringae*, *Pseudomonas fluorescens*, and *Legionella pneumophila*. *L. pneumophila* exhibits a form of surface translocation that is the most similar to sliding motility and is dependent on the T2SS [83]. The second group of sliding bacteria is that which requires additional secreted components such as exopolysaccharides. This type of movement is characteristic of *B. subtilis* and *Sinorhizobium meliloti* species. The bacteria belonging to the third group require a factor other than a surfactant component for sliding, and this motility occurs in *S. enterica* serovar Typhimurium and *Mycobacterium smegmatis* species [84]. A recent study showed that *S. enterica* serovar Typhimurium exhibits sliding motility under magnesium-limited conditions, which induces PagM protein secretion [30]. *S. aureus*, common in wound and skin infections, is historically defined as non-motile, but recent studies proved the passive motion abilities of this species. This passive motion was termed spreading and was distinguished from sliding by the production of specific phenol-soluble modulin (PSM) surfactants. *S. aureus* spreading manifests as a finger-like dendrite that emerges from the central colony on a solid agar medium. The spreading is followed by the formation of a comet-like structure, which first emerges from the central colony, and the comet heads are composed of aggregated cells joined together through a matrix of slime. PSM production is considered to be a virulence factor in *S. aureus* [85].

## 3. Conclusions

Skin and wound infections are serious and unsolved medical problems. Bacterial translocation favours recurrent infections, and bacterial motility is considered to be a major virulence factor, where the relationship between bacterial migration and chemotaxis enables bacteria to sense the source of nutrients and subsequently translocate into the niches for colonisation [48]. Most bacteria display at least one form of motility, either involving appendages (swimming, swarming, and twitching motilities) or without appendages (gliding and sliding motilities). Bacterial movement involves a number of molecular mechanisms that could serve as targets for antibacterial therapy during skin and wound treatment, enhancing the action of antibacterial and/or disinfectant agents. The main bacterial factors involved in bacterial motility are appendages such as flagella and pili, and recent studies indicate that bacterial type IV pili are a promising therapeutic target. Moreover, it was shown that hydrogel coatings can inhibit swarming and swimming motilities by distorting *E. coli* flagella functions, which leads to the inhibition of biofilm formation. By contrast, appendage-less bacteria need to produce large amounts of lubricants to reduce the frictional force, which could also serve as a useful target for antibacterial therapies. In addition, the chemical modification of the interaction between the bacterial cell and the substrate may affect the gliding motility by disrupting the interaction of focal adhesion sites (FACs) with the substrate, thereby inhibiting the movement of bacteria and facilitating its eradication.

## Figures and Tables

**Figure 1 ijms-24-01707-f001:**
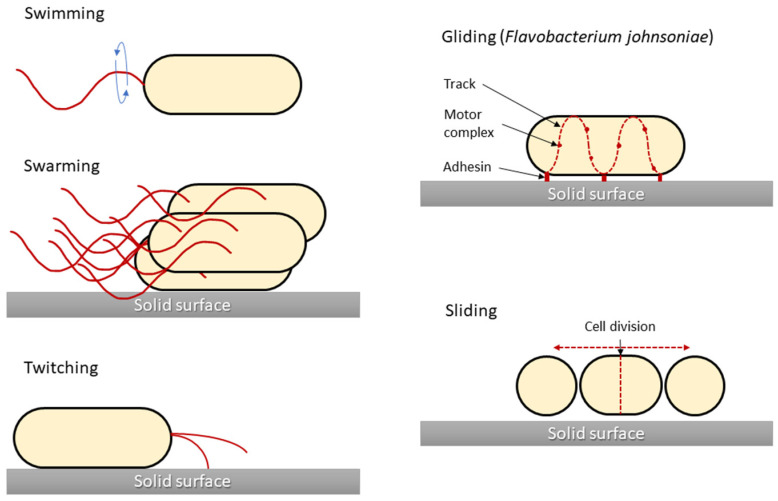
Graphical presentation of five bacterial movement types. Structures responsible for generating bacterial movement are marked in red. Sliding motility is presented for *Flavobacterium johnsoniae* as a model for this type of motility, although for other bacteria (i.e., *Myxococcus xanthus*) there are differences in the cell propulsion mechanism. The figure was prepared based on the data and figures presented in [31].

**Table 1 ijms-24-01707-t001:** Microbial states of the infected wound [18,19].

State	Microbial Activity and Host Response	Recommended Action
Contamination	Microbes enter the wound but there is no host response.	Observation
Colonisation	Microorganisms multiply without causing wound deterioration with only a weak host immune response.	
Critical colonisation	A large number of bacteria cause a delay in the wound healing process with a poor immune response of the host.	Intervention
Infection	A vast number of bacteria lead to delays in wound healing and trigger an often symptomatic inflammatory response caused by a strong immunological host response.	

## Data Availability

Not applicable.

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
