# Peer review of "Bacterial Motility and Its Role in Skin and Wound Infections"

_ijms, 2023, doi:10.3390/ijms24021707_

Round 1

Reviewer 1 Report

1.      In my opinion, the abstract should be improved. Authors should focus more on the purpose of the review.

2.      The main aim of this work should be clearly given in the last paragraph of the introduction (see Lines 85 – 85, in my opinion the aim of this work should be given).

3.      Table 1 -Is this a copy of work 17?

4.      Line 73 – The usage of ultrasound and electromagnetic therapy should be detailed described.

5.      Line 74-74 – Is there any alternative methods?

6.      Line 77 – These methods should be detailed described.

7.      The main of this work is “bacterial motility and its role in skin and wound infections”. The main boy of this paper is connected with ttypes of bacterial movement”. I think that the conclusion should be connected with the main topic of this work. This information should bed added.

Author Response

Dear Reviewer,

Thank you for your valuable comments. We do correct the manuscript according your suggestions and we’ve addressed all of them. For the first of all, we’ve rebuilt the abstract section as you suggest. I hope this modification will simplify this manuscript and make it easier to read. The following improvements are listed below.

Thank you for your valuable comments. We reviewed the manuscript according to your suggestions and addressed all of them. First of all, we rebuilt the summary section as you suggested. I hope this modification will make the manuscript easier to read. The following improvements are listed below.

The Reviewers’ comments and authors’ answers:

Reviewer 1

  1. In my opinion, the abstract should be improved.Authors should focus more on the purpose of the review.

Response 1: Corrected abstract is submitted.

  1. The main aim of this work should be clearly given in the last paragraph of the introduction (see Lines 85 – 85, in my opinion the aim of this work should be given).

Response 2: Indeed, aims of manuscript need to be in end of introduction (line 104-110), we have added appropriate sentences”

  1. Table 1 - Is this a copy of work 17?

Response 3:

Table 1 was in Polish language. We do modification,  translation and combination of sources of table. Table  on  based on references: [18] JawieÅ„, A.; Bartoszewicz, M.; Przondo-Mordarska, A.; Szewczyk, M.T.; Kaszuba, A.; Urbanek, T.; Staszkiewicz, W.; Sopata, M.; Kucharzewski, M.; Korzon-Burakowska, A.; et al. Wytyczne PostÄ™powania Miejscowego i Ogólnego w Ranach ObjÄ™tych Procesem Infekcji. Leczenie Ran 2012, 9, 59–75 and The information for this table was partially taken from article: [19] White, R.; Cutting, K. Critical Colonisation of Chronic Wounds: Microbial Mechanisms. Wounds 2008, 4, 70–78.

Both articles were added to the manuscript .

  1. Line 73 – The usage of ultrasound and electromagnetic therapy should be detailed described.
  2. Line 74-74 – Is there any alternative methods?
  3. Line 77 – These methods should be detailed described.

Response 3,4, and 5: This section was rewritten and the descriptions of the therapies were added. We believe that extending this section to other therapies diverts this manuscript from its main purpose of reviewing the types of bacterial motility in the context of wound infection.

  1. The main of this work is “bacterial motility and its role in skin and wound infections”. The main boy of this paper is connected with ttypes of bacterial movement”. I think that the conclusion should be connected with the main topic of this work. This information should bed added.

Response 6:

Indeed wound skin infection and bacterial motilities are not obviously connected. However, bacterial cells motilities is one several pathogenic factors that might facilitate spread of pathogenic bacteria over new skin area. New part of statements were added to Conclusion part.

Reviewer 2 Report

The authors are presenting a review article regarding bacterial motility and its role in skin and wound infections. The article is well presented, however I have some critical points that I feel needs to be corrected before this manuscript can be accepted for publication.

Below are my specific comments and suggestions for improvement.

Throughout the manuscript, please correct the line breaks so that they do not occur in the middle of words.

Row 11, I suggest to write “….between individuals…” instead of “…in induviduals…”.

Row 14, I suggest to write “… difficult to treat…” instead of “…difficult to cure…”  

Row 44, I suggest to write “….with varied clinical development…”, or “….with varied clinical appearance…”  instead of “…with a varied clinical presentation…”.

Row 264, I suggest to write “…development [63]…” or similar instead of “…development[63]…”.

Row 145-155, 252-256, 286-300, Clarify which publications that presents this research.

Author Response

Dear Reviewer,

Thank you for your valuable comments. We reviewed the manuscript according to your suggestions and addressed all of them. First of all, we rebuilt the summary section as you suggested. I hope this modification will make the manuscript easier to read. The following improvements are listed below.

The Reviewers’ comments and authors’ answers:

Answers to Reviewer 2

Thank you for your valuable comments. We reviewed the manuscript according to your suggestions and addressed all of them.

The authors are presenting a review article regarding bacterial motility and its role in skin and wound infections. The article is well presented, however I have some critical points that I feel needs to be corrected before this manuscript can be accepted for publication.

Below are my specific comments and suggestions for improvement.

  1. Throughout the manuscript, please correct the line breaks so that they do not occur in the middle of words.

Response 1: Line breaking is imposed by the MDPI form, which we filled in with the previously prepared text. Looking through other articles from this journal, I noticed that this way of formatting the text is typical for the journal IJMS.

  1. Row 11, I suggest to write “….between individuals…” instead of “…in induviduals…”.
  2. Row 14, I suggest to write “… difficult to treat…” instead of “…difficult to cure…”  

Response 2 and 3: corrected, whole abstract was rewritten.

  1. Row 44, I suggest to write “….with varied clinical development…”, or “….with varied clinical appearance…”  instead of “…with a varied clinical presentation…”.

Response 4: corrected to” varied clinical appearance”

  1. Row 264, I suggest to write “…development [63]…” or similar instead of “…development[63]…”.

Response 5: corrected

  1. Row 145-155, 252-256, 286-300, Clarify which publications that presents this research.

Response 6: Appropriate references were added in mentioned sections.

Round 2

Reviewer 1 Report

The authors took into account all my comments in the revised version of the paper. The work may be published in the presented form.